# Industrial Informatics: Emerging Trends and Applications in the Era of Big Data and AI

**Mirjana Pejić Bach** [1,*] **, Arian Ivec** [2] **and Danijela Hrman** [3]

1 Department of Informatics, Faculty of Economics and Business, University of Zagreb, 10000 Zagreb, Croatia
2 Department of Physics, Faculty of Science, University of Zagreb, 10000 Zagreb, Croatia
3 Probotica d.o.o., 10000 Zagreb, Croatia
* Correspondence: mpejic@net.efzg.hr; Tel.: +385-1-238-3333

**Abstract:** Industrial informatics is a rapidly developing scientific field that deals with the knowledge-based automation of industrial design and manufacturing processes. In the last decade, industrial informatics has been strongly influenced by the rapid rise of data-based technologies such as Data Science, Big Data, and artificial intelligence. The goal of this paper is to provide a literature review of academic research analyzing the extensive spectrum of industrial informatics. Articles indexed in Scopus with the term "Industrial Informatics" in the title, abstract, or keywords were extracted since the term emerged in the 1990s, over a period of 29 years. The main journals, conferences, authors and countries were studied using bibliometric analysis. Text mining using VosViewer was used to extract the thematic groups of research related to industrial informatics, which are as follows: (i) Internet of Things, (ii) machine learning, (iii) engineering education, (iv) cyber–physical systems, and (v) embedded systems. We also found that China, Germany, and Brazil dominate research in industrial computing. The results showed that research in industrial informatics is related to the emergence of new methods and tools, and is nowadays shifting towards the application of intelligent methods such as machine learning and Big Data.

**Keywords:** industrial informatics; Scopus; bibliometrics; VosViewer; Big Data; AI



## 1. Introduction

Industrial informatics is a scientific field that deals with the research, design, use and innovation of information and communication technologies (ICT) in industrial applications [1]. The opportunities offered by ICTs have transformed industrial and business processes by changing the established paradigms of industrial practice. These ICTs can enable the development of new types of industrial engineering [2]. It involves the integration of information, control, and communication systems to improve industrial efficiency, production, and quality environments, as well as to develop, deploy, and control advanced manufacturing systems and processes. Industrial informatics builds on a discourse of ICT design, innovation, and use formulated in the computer science discipline [1,3]. Industrial informatics is a focused effort that represents a dedicated science aimed at understanding and improving ICT-related design, use, and innovation processes [4] to solve increasingly complex interactions [4–7].

Industrial informatics aims at ICT-based factory automation to improve manufacturing processes, products and their underlying systems. The benefits of industrial informatics lie in the integration of ICT into data acquisition, processing, and leveraging it to increase the efficiency, effectiveness, reliability, and sustainability of enterprises. Due to its multidisciplinary nature, industrial informatics research has attracted the attention of a wide range of stakeholders [8]. The scope of industrial informatics has expanded significantly, from a supporting role in the design of manufacturing processes to its use throughout the product life cycle, including reuse, remanufacturing, maintenance, recycling, and outsourcing [8]. New approaches to networked cyber–physical systems that enable close monitoring and

synchronization between physically connected systems and the cyber–computer space have been developed. The approach to creating and implementing the framework for such networked systems differs for each monitored physical system. Systematic deployment of cyber–physical systems enables a network of machines to operate more efficiently, collaboratively, and resiliently. Such changes can potentially catapult the industry to the next stage of evolution known as Industry 4.0. [9]. Artificial intelligence (AI) is the main driving force behind the evolution of industrial computing, with varying degrees of interest and success. New advances in computing infrastructure, algorithmic innovations, and the large increase in available industrial data due to the increasing digitization of interactions, processes, and systems have dramatically increased the use of AI technologies. The ability to use AI for real-time control and monitoring will be the main generator of competitive advantage from now on. The ultimate goal is to construct fully autonomous systems that can meet not only narrow organizational, but also social, economic, and environmental goals [10].

Very few efforts have been made to evaluate industrial computing research using bibliometric or scientometric studies, although industrial computing research has attracted considerable attention in recent decades [11]. Newer bibliometric techniques, such as scientific or bibliometric mapping techniques, have gained traction in providing new insights into the specifics of the development of a scientific discipline and identifying patterns or groupings in bibliometric datasets. This has been achieved by identifying the links between different scientific publications [12]. In particular, science mapping represents how disciplines, subject areas, publications, and authors are interconnected. Its goal is to highlight the complex and dynamic features of scientific investigations, and this is achieved by showing these relationships [13]. These mappings demonstrate the interactions between different queries, i.e., authors, articles, publications, and keywords, and thus are usually created based on citations, co-citations, bibliographic linkage data, associations, and co-occurrence of keywords. These methods have been used in a variety of research areas, including Big Data [14], blockchain [15], machine learning [16], and artificial intelligence in sports [17]. However, the scientific mapping method has never been used to conduct research analysis in industrial informatics. Since the term industrial informatics was coined in the 1990s, there are already a large number of published research papers on industrial informatics. This means that our approach is an alternative solution and an effective way to show the evolution and structure of studies conducted in industrial informatics.

In analyzing keywords in industrial informatics publications, we applied a sophisticated visualization method to detect clusters of related keywords. In addition, we performed a citation analysis to identify the citable phrases represented in the bibliometric maps. To be more specific, our objectives were the following:

i. to discover patterns in the results of scientific publications and the total number of articles published by journals, nations, and research institutions,
ii. to provide a comprehensive overview of the field by visually depicting the major areas of study, their relationships, and their evolution,
iii. to identify the phrases cited most often.

To achieve these goals, Elsevier's Scopus database [18,19] was used to extract scientific articles related to industrial computing. In this article, the studies conducted in the field of industrial informatics are visualized. This will provide the research community working in industrial informatics with valuable tools to update their achievements in this field.

For a detailed overview of the methodology, see the Materials and Methods section. In the Results section, we show term and term citation maps of the analyzed periods and statistical overviews of the bibliometric data. In the Discussion section, we analyze the results and discuss their implications. Finally, in the Conclusion section, we summarize our main findings and discuss further research opportunities.

## 2. Materials and Methods

### 2.1. Data Gathering

We used the Scopus database to collect bibliographic information on publications in the field of industrial computing from 1950 to 2023. Scopus was chosen for its consistency compared to other databases [20,21]. To find relevant publications in the field of industrial informatics, we used the keywords listed in Table 1 in the Title, Abstract, and Keywords fields. The search approach we used has the following limitations:

(i)   The dataset contains only articles and reviews; books and proceedings were not included;
(ii)  We have considered only English-language publications;
(iii) Articles not indexed in the Scopus database were omitted due to the nature of our search.

**Table 1.** Bibliometric search strategy and results.

| Database | Period | Search Term | # of Documents |
|---|---|---|---|
| Elsevier Scopus | 1950–2023 | TITLE-ABS-KEY ("Industrial Informatics") | 1077 documents |
| Elsevier Scopus | 1950–2023 | TITLE-ABS-KEY ("Industrial Informatics") AND (EXCLUDE (DOCTYPE, "ed") OR EXCLUDE (DOCTYPE, "tb") OR EXCLUDE (DOCTYPE, "bk") OR EXCLUDE (DOCTYPE, "ch") OR EXCLUDE (DOCTYPE, "er") OR EXCLUDE (DOCTYPE, "no")) | 1045 documents |

As Table 1 shows, constraint #2 has no significant impact on the results, since the exclusion of various categories (Editorial, Retracted, Book, Book Chapter, Erratum, Note) results in only 32 documents being discarded, which is less than 1 percent of the documents examined.

### 2.2. Bibliometric Mapping and Clustering

To create a comprehensive picture of industrial informatics, we used bibliometric maps [22,23] created with VOSviewer software, which specializes in quantifying and visualizing scientific bibliometric maps [24]. We created term maps [25], two-dimensional representations of a scientific field in which strongly related terms are close to each other and less strongly related terms are further away [26]. Term maps provide an overview that can be used to identify and classify the structure of a scientific field [27].

Following the language processing algorithms of VOSviewer [24], we examined all keywords found in the titles and abstracts of the articles using the VOS mapping approach [28,29]. The VOS mapping determines the best position for a given term on the map by minimizing the weighted sum of squared Euclidean distances between all pairs of elements. This mapping approach allows words to be placed on the map so that the distance between each pair visually indicates their similarity. For any two terms indexed by $i$ and $j$, the similarities between the terms are based on the number of common occurrences in the title or abstract of the same publication according to the following equation:

$$\text{similarity}_{i,j} = \frac{c_{ij}}{c_i c_j},$$  (1)

where $c_{ij}$ is the number of publications in which the two terms $i$ and $j$ appear together, while $c_i$ and $c_j$ are the number of publications in which each term appears [29]. The more publications in which two terms appear, the more strongly the terms are related. Terms that frequently appear in the same publications are grouped in a term map, while weakly related terms (low common occurrence) are further apart. Each phrase is represented by a circle, with the diameter and size of the label indicating the number of publications containing

the term in question in the title or abstract. The software uses the VOS clustering approach to identify clusters of similar phrases, a weighted and parameterized form of modularity-based clustering [28,30–33]. A cluster can be viewed as a research topic containing one or more research topics whose color is determined by the average citation impact of the articles in which the term appears [34,35].

The color assigned to a term in the Term Citation Maps reflects its average citation impact. To reduce bias due to a paper's year of publication, the number of citations received is divided by the average number of citations for all articles published in that year. A publication's normalized citation value ranges from 0 to 2. Colors are assigned based on these values. The colors vary from blue (average 0), corresponding to a low average citation impact, to green (average 1), corresponding to a high average citation impact, to red (average above 2), with blue and red representing low and high citation impact, respectively [36]. All maps shown can be freely explored with VOSViewer using files S1–S4; see Supplementary Materials for details.

Term maps, then, provide only a simplified picture of reality, which can lead to loss of information and thus do not paint a comprehensive picture of the area under study. This limitation should be kept in mind when analyzing a concept map. Dividing data into different time periods is the typical strategy to study changes in a scientific field over time [37].

We divided the retrieved bibliographic data into four periods: 1994–2007, 2008–2013, 2014–2018, and 2019–2023. Note that the first period is larger because industrial informatics was still nascent during this period, i.e., few articles were published and merged with the previous period. Term maps and citation maps were created for each period and can be found in Tables S1–S4.

## 3. Results

### 3.1. Broad Publication Trends

We retrieved 1045 publications of various types from the Scopus database spanning 29 years. The largest number were conference papers (97%), while the rest consisted of articles.

Figure 1 shows the publication frequency of industrial informatics from 1994 to 2023. Until 2012, the number of publications per year was at most in the single digits, after which there was a sharp increase in publications, with a peak in 2019. A total of 159 journals were published in this field.

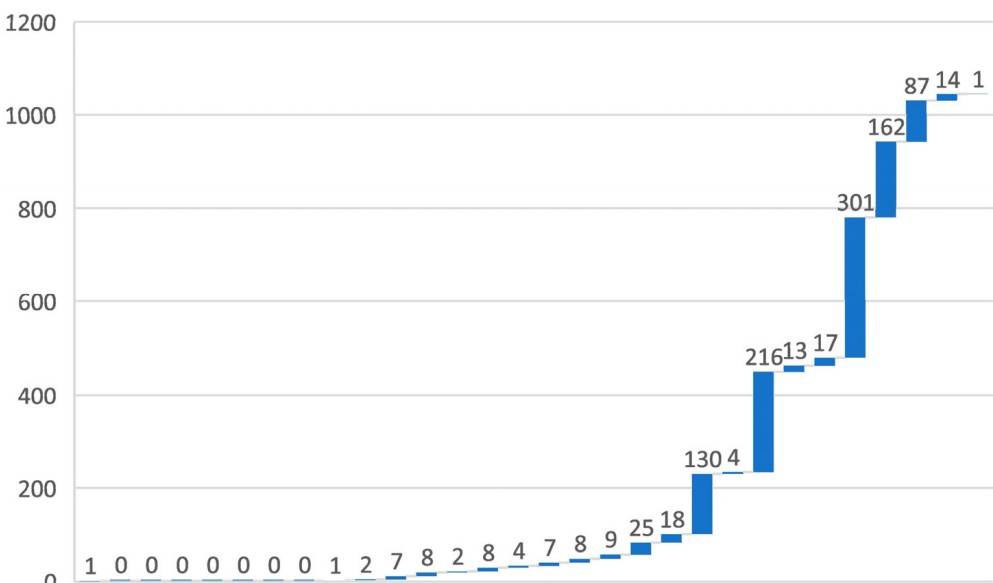

**Figure 1.** *Cont.*

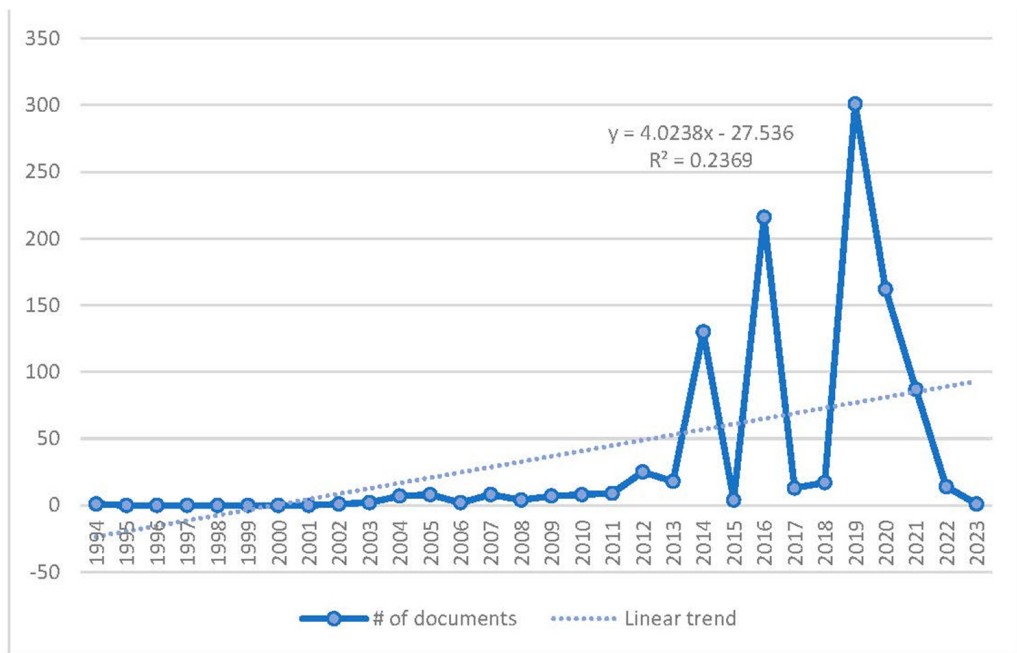

**Figure 1.** Industrial informatics publication trends from 1994 to 2013. The upper figure shows the yearly increase in publications in industrial informatics. The lower figure shows the absolute number of published papers in industrial informatics. The straight line is the linear fit of the data, with the parameters given in the figure.

Table 2 shows the top journals in information systems throughout the observation period. The most active journal was *IEEE Transactions on Industrial Informatics*.

**Table 2.** Most active journals and conferences in industrial informatics from 1994 to 2023. Source: the authors' work.

| Title | # of Documents |
|---|---|
| *IEEE Transactions on Industrial Informatics* | 46 |
| *Applied Mechanics and Materials* | 8 |
| *IECON Proceedings Industrial Electronics Conference* | 8 |
| *IFAC Proceedings Volumes IFAC Papers online* | 7 |
| *Advances In Intelligent Systems and Computing* | 6 |
| *IEEE Access* | 3 |
| *IEEE Transactions on Industrial Electronics* | 3 |
| *Riai Revista Iberoamericana de Automatica e Informatica Industrial* | 3 |
| *ACM International Conference Proceeding Series* | 2 |
| *Applied Sciences Switzerland* | 2 |
| *Enterprise Information Systems* | 2 |
| *IEEE Industrial Electronics Magazine* | 2 |
| *IEEE Internet of Things Journal* | 2 |
| *IEEE Transactions on Education* | 2 |
| *Lecture Notes in Computer Science* | 2 |
| *Revista Iberoamericana de Tecnologias del Aprendizaje* | 2 |

In Figure 2, We show the breakdown of publications by country, with China clearly leading in the number of publications in industrial computing. These countries have the most prolific institutions publishing research on industrial informatics (Table 3).

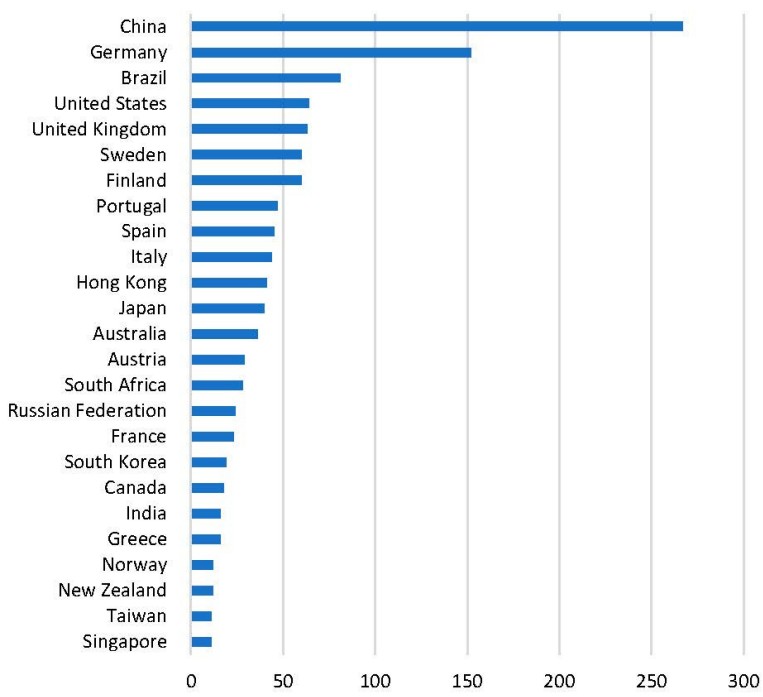

**Figure 2.** Countries that produced at least 10 publications in industrial informatics.

**Table 3.** Affiliations with more than 10 publications in industrial informatics. Source: the authors' work.

| Affiliation | # of Documents |
|---|---|
| Aalto University | 36 |
| Luleå University of Technology | 34 |
| City University of Hong Kong | 27 |
| Technical University of Munich | 27 |
| University of Pretoria | 22 |
| Chinese Academy of Sciences | 20 |
| Universidade Federal do Rio Grande do Sul | 18 |
| Shanghai Jiao Tong University | 17 |
| Universidade do Porto | 17 |
| Old Dominion University | 16 |
| Universidade Federal de Santa Catarina | 15 |
| Wuhan University of Technology | 14 |
| Instituto Politecnico de Braganca | 14 |
| Technische Universität Wien | 14 |
| Rheinisch-Westfälische Technische Hochschule Aachen | 13 |
| Consiglio Nazionale delle Ricerche | 13 |
| Beihang University | 12 |
| Tampere University | 12 |
| Institute for Systems and Computer Engineering, Technology, and Science | 11 |
| Universidade Federal do Rio Grande | 11 |

Most publications in the field of information systems are in the area of computer science and engineering (Figure 3). However, a non-negligible number of publications come from the field of mathematics-related decision sciences.

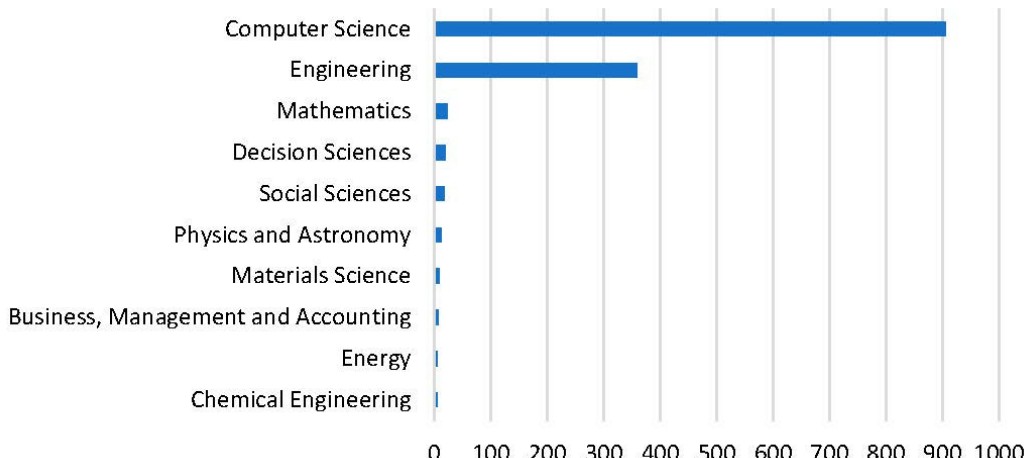

**Figure 3.** Subject areas with more than 10 publications in industrial informatics.

The analysis of the frequency of occurrence of keywords and terms can be seen in Figure 4. Industrial informatics, of course, dominates all other terms and occurs more than 10 times as often as the second most frequently mentioned term. Among the remaining terms, it is interesting to note, as will become clear later in the temporal analysis, that newer terms are at the top of the list, which is typical of a field that is growing as rapidly as industrial computing.

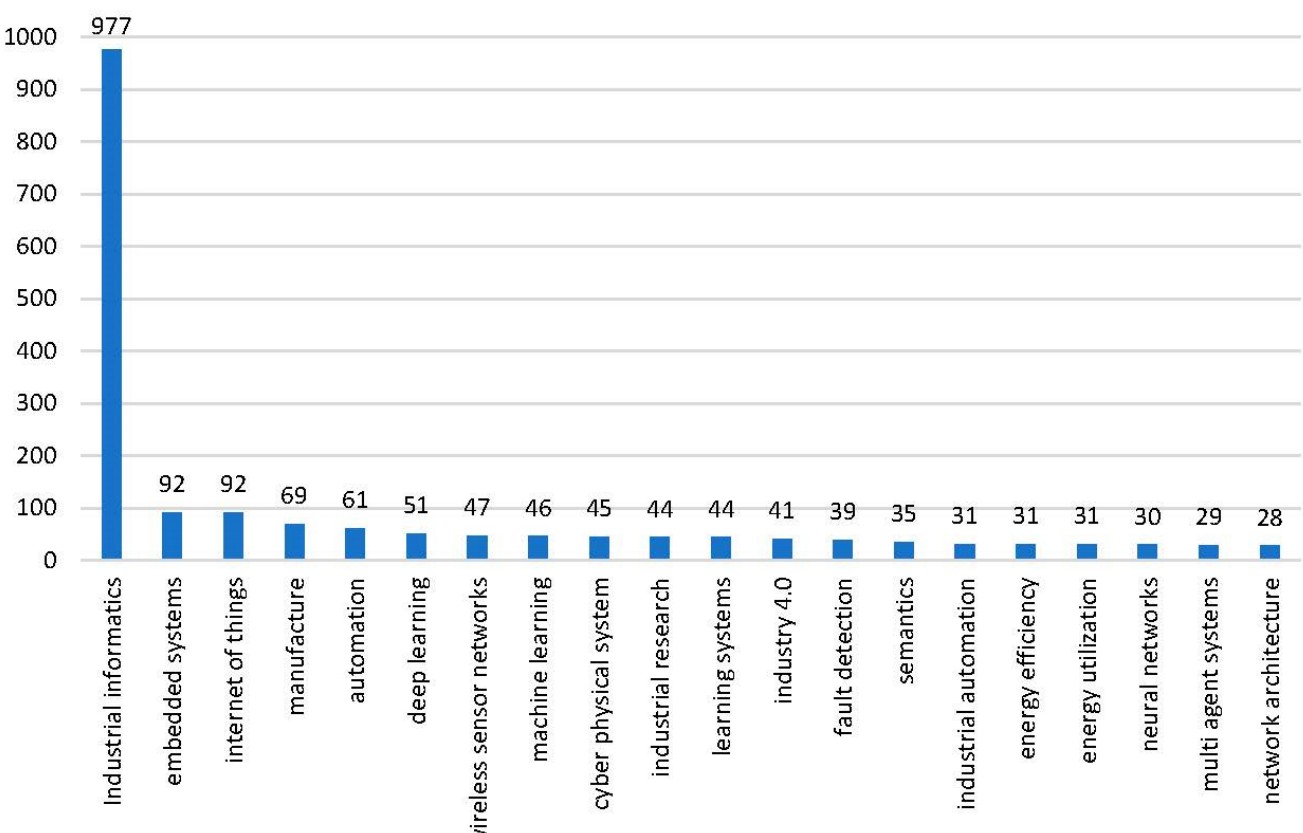

**Figure 4.** The 20 most often occurring terms (1994–2023). Note: industrial informatics was not excluded.

### 3.2. Evolution of Research Topics and Their Citation Impacts

Figure 5 shows the most common terms and the average year of publication. We can see that the terms with the most occurrences are found in the last eight years, which shows that terminology is evolving rapidly, as is the state of research itself.

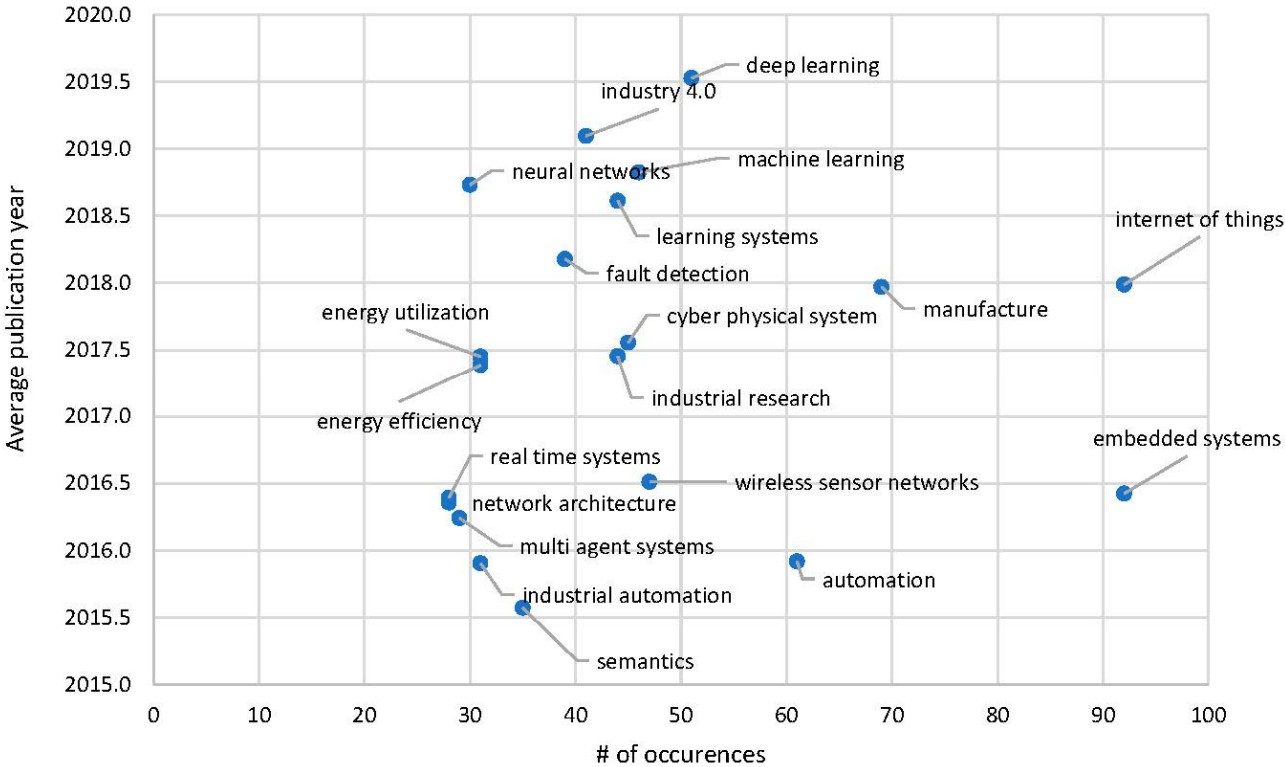

**Figure 5.** The occurrence and average publication year of the most frequent terms. Note: industrial informatics was not excluded.

In Figures 6–13, we show the term and citation maps for each time period for four different time periods since 1994. Colors are used in the term maps to highlight clusters of related phrases, while the colors in the citation maps reflect the average citation effect across all publications containing the term.

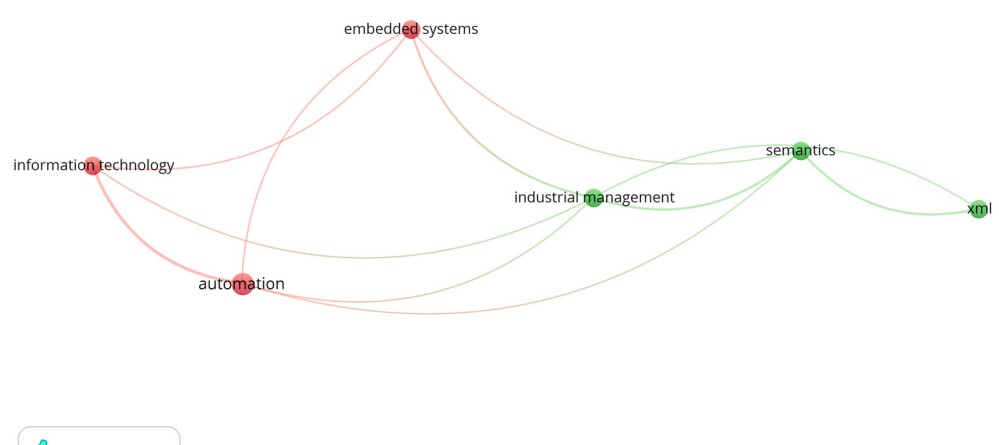

**Figure 6.** Term map constructed from industrial informatics publication data from 1994 to 2007. The size of a term's circle shows the citation number, while its color shows the cluster it belongs to. The lines are co-occurrence links between terms.

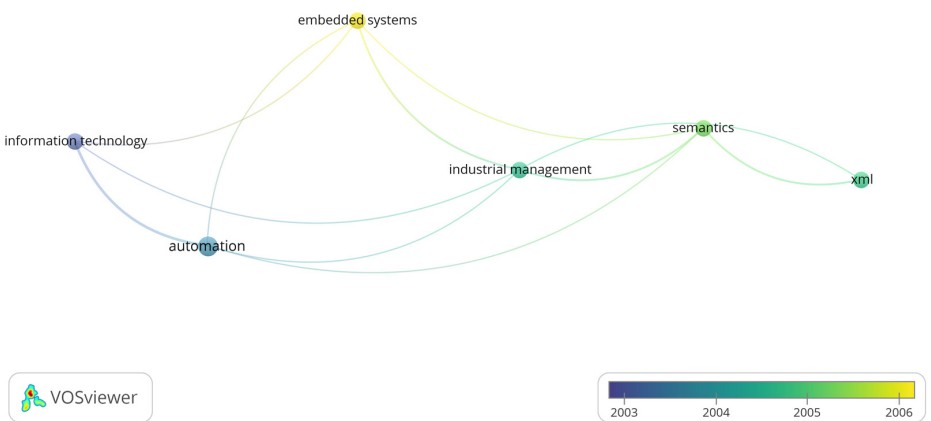

**Figure 7.** Term citation map constructed from industrial informatics publication data from 1994 to 2007. The size of a term's circle shows the number of citations, and the lines are co-occurrence links between terms. Colors show the average publication year for both links and circles.

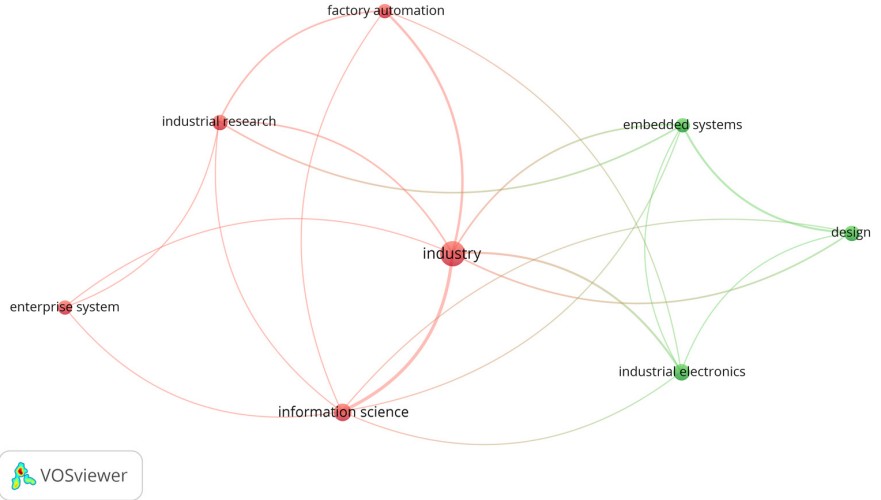

**Figure 8.** Term map constructed from industrial informatics publication data from 2008 to 2013. The size of a term's circle shows the number of citations, while its color shows the cluster it belongs to. The lines are co-occurrence links between terms.

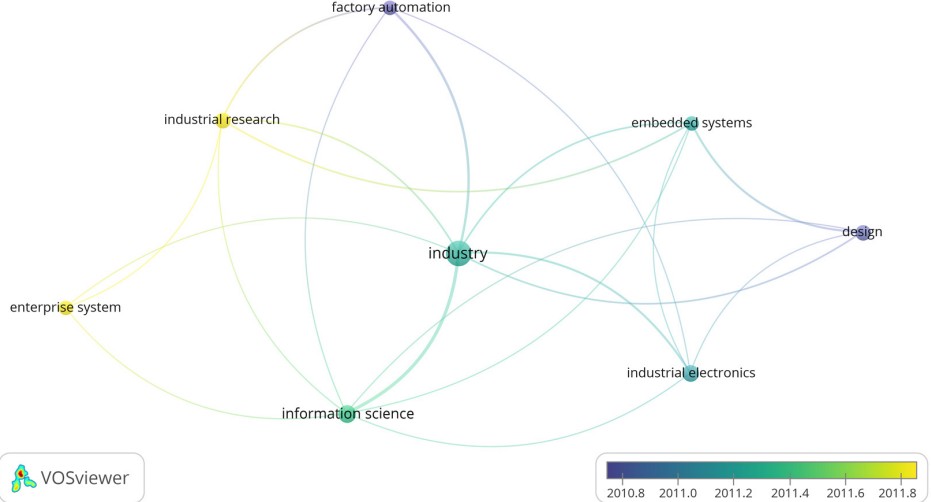

**Figure 9.** Term citation map constructed from industrial informatics publication data from 2008 to 2013. The size of a term's circle shows the number of citations, and the lines are co-occurrence links between terms. Colors show the average publication year for both links and circles.

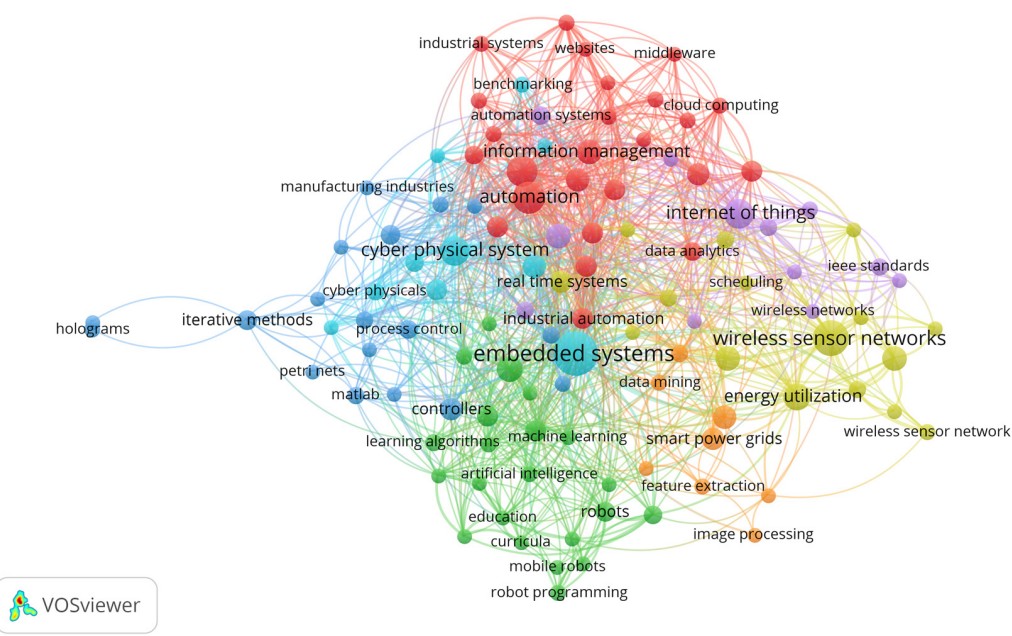

**Figure 10.** Term map based on industrial informatics publication data from 2014 to 2018. The size of a term's circle shows the number of citations, while its color shows the cluster it belongs to. The lines are co-occurrence links between terms.

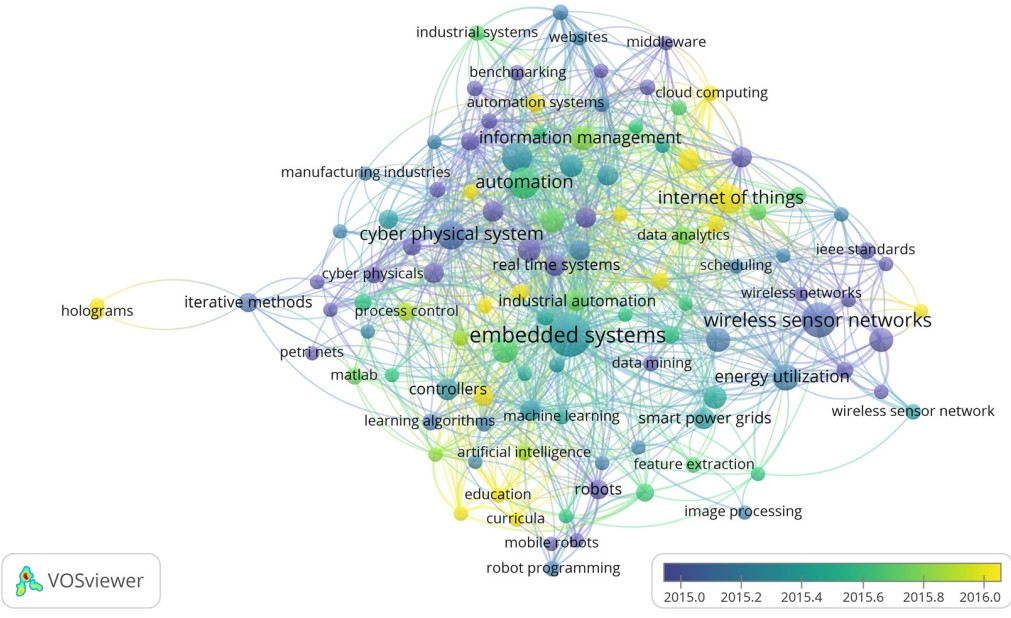

**Figure 11.** Term citation map constructed from industrial informatics publication data from 2014 to 2018. The size of a term's circle shows the number of citations, and the lines are co-occurrence links between terms. Colors show the average publication year for both links and circles.

Figure 6 shows the map of terms we created for 1994–2007. The six terms shown on the map are divided into two groups (red and green). Automation, embedded systems, and information technology are in the red cluster on the left, while the green cluster on the right consists of industrial management, semantics, and XML.

The associated map of term mentions in Figure 7 shows how small the field of industrial computing is in this period and that no term is much more represented than its neighbors.

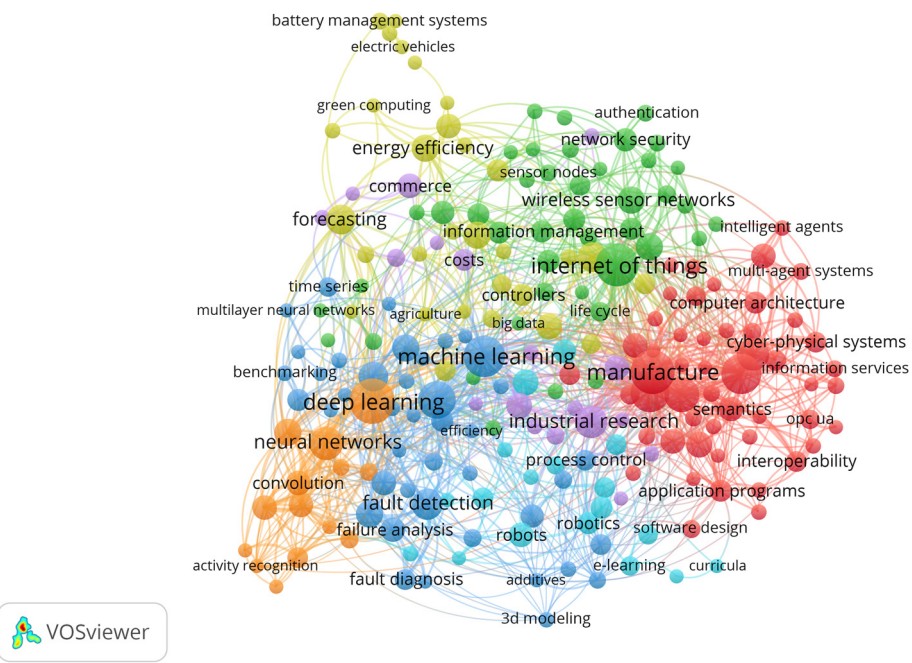

**Figure 12.** Term map constructed from industrial informatics publication data from 2019 to 2023. The size of a term's circle shows the number of citations, while its color shows the cluster it belongs to. The lines are co-occurrence links between terms.

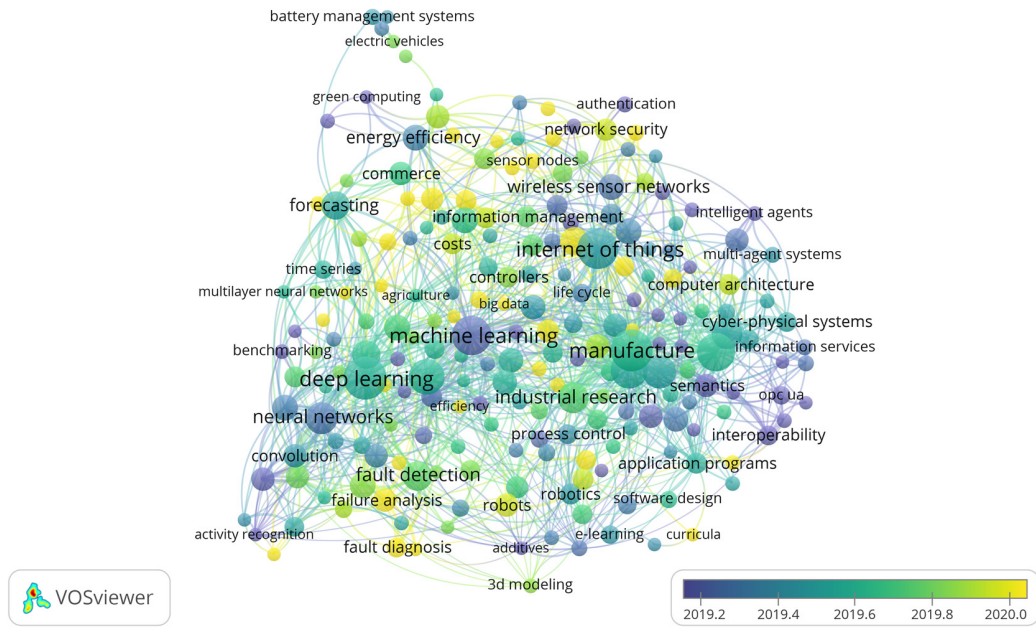

**Figure 13.** Term citation map constructed from industrial informatics publication data from 2019 to 2023. The size of a term's circle shows the number of citations, and the lines are co-occurrence links between terms. Colors show the average publication year for both links and circles.

Figure 8 shows the enlargement of the field. However, the terms are still grouped into only two clusters. The appearance of the central industry term, which refers to all terms except design, suggests that the field is becoming a coherent whole.

In Figure 9, which shows the map of term mentions, the yellow-colored terms from industrial research and enterprise systems are the most recent, indicating that the state of the field is moving toward broad application.

As can be seen in Figure 10, a rapid increase in terms and links can be observed in the next period. Different clusters have been formed, the central term being embedded

systems, i.e., industrial computing functions that can run autonomously from industrial systems themselves [38–40].

The map of term mentions is shown in Figure 11 and the most recent additions to the field can be seen. On the lower right, recent terms related to students and education have appeared, indicating the maturity of the field and the need to train researchers due to the increasing complexity of the field. The rapid emergence of the Internet of Things [41,42] and Big Data can be seen [43].

In Figure 12, the latest concept map, which extends to the present day, shows the dominance of the Internet of Things. We can identify several significant clusters, of which the purple and yellow clusters are largely concerned with AI [44,45], whose use in industrial computing has increased rapidly due to the increasing availability of computing resources in both embedded and non-embedded systems [46,47] and the use of cloud computing [48].

A complete listing of the VOSviewer settings used can be found in Table 4, which shows that the average publication year of each cluster is much closer to the end of each period, indicating that research in industrial computing is growing rapidly.

**Table 4.** Bibliometric mapping and clustering summary. Source: the authors' work.

| Period | 1994–2007 | 2008–2013 | 2014–2018 | 2019–2023 |
|---|---|---|---|---|
| Type of analysis | Co-occurrence | Co-occurrence | Co-occurrence | Co-occurrence |
| Unit of analysis | Keywords | Keywords | Keywords | Keywords |
| Counting method | Full count | Full count | Full count | Full count |
| Minimum threshold value | 5 | 5 | 5 | 5 |
| Extracted keywords | 288 | 712 | 3671 | 5968 |
| Extracted keywords that occur more than the threshold | 7 | 9 | 112 | 216 |
| Excluded term | Industrial Informatics | Industrial Informatics | Industrial Informatics | Industrial Informatics |
| Occurrence of excluded term | 17 | 51 | 362 | 547 |
| Minimum cluster size | 3 keywords | 3 keywords | 4 keywords | 5 keywords |
| # of clusters | 2 | 2 | 7 | 7 |
| # of keywords | 6 | 8 | 111 | 215 |
| Links | 11 | 20 | 1057 | 3254 |
| Total link strengths | 17 | 38 | 1443 | 4739 |

Source: the authors' works.

## 4. Discussion

According to our analysis, only 29 English-language publications were published in industrial informatics between 1994 and 2007, with the earliest publication also indexed by Scopus only in 1994. The number of scientific publications that our query yields depends on the keywords used in the articles. It is not a perfect representation of the state of the field at any given time [49]. This is especially true for the earliest period from 1994 to 2007, when the field of information systems was just emerging. An emerging field does not yet have a standardized terminology, and a retrospective search of this period using the terminology in use today introduces survivorship bias [50]. This problem can artificially lower the measured publication activity and therefore must be considered when interpreting the data.

From 1994 to 2007, the number of publications increased rapidly, peaking in 2019. A comparable rapid increase in publication numbers has been reported in similar areas

of network ecology [51,52], engineering research [53], and global climate change [54–56]. The lack of comparable bibliometric studies of information systems makes it difficult to validate our results, but also underscores the urgent need for this study and further analysis of the field. Our study analyzed 1045 papers published in 56 different journals. In addition, 100 articles were published in the 17 major journals, *IEEE Transactions on Industrial Informatics* being the most prolific journal with 46 articles published. *Applied Mechanics and Materials* and *IECON Proceedings* ranked second and third, respectively. These results illustrate the multidisciplinary nature of industrial informatics research, which is due to the continued interest of various government and private stakeholders in advancing the field, both from scientific and business perspective.

This research is of global interest, as 68 different countries contributed with at least one publication. The top five countries, where the principles of industrial computing are used extensively, are China, Germany, Brazil, the United States, and the United Kingdom, with Brazil being the outlier rising above given the level of research and industrial development compared to the other countries in the top five [57]. China's dominance is not surprising and is largely in line with trends in other fields. The field is also very top-heavy, with the top five responsible for nearly 45% of all published research and the top half of countries responsible for 92% of all published work.

Using the scientific mapping approach, we created a series of maps showing clusters of co-occurring words in which we identified different study topics. The average year of link publication was also used to color the keywords from blue to yellow. From the coloring, we can see that the field expands rapidly in each time period, as the average publication year is much closer to the end year of each time period than the beginning year, and yellow hues dominate the corresponding term citation maps. In each term citation map, obsolete branches are marked in blue. These branches tend to become less relevant and even extinct in subsequent periods. We also evaluated the chronological evolution of industrial informatics research over the last 29 years, as the mapping approach was used at different times.

The absence of terms on the map and their ambiguity in 1994–2007 and 2008–2013 hindered the grouping of specific research areas. Narrower terms have emerged, leading to more precise definitions of clusters and research areas. By comparing maps across study periods, we can observe the emergence, disappearance, and repositioning of terms in individual clusters, providing valuable insights into the dynamics of research topics in a given area map. For example, wireless sensor networks were closely related to topology in 2014–2018, but the rapid emergence of the Internet of Things in the next period, 2019–2023, catapulted it into its cluster. The emergence of new terms is increasing the size of the clusters, which means that research in industrial informatics is becoming more complex, with subfields and subspecialties emerging in research and the industry, respectively.

Special attention should be paid to the most recent map, i.e., 2019–2023, as it represents the current state of research in industrial computing. The Internet of Things, machine learning, neural networks, and automation are the main features of today's industrial informatics. Our results show that neural networks and their associated methods for creating artificial intelligence have decoupled from more traditional machine learning methods and formed their own cluster, shown in purple. This shows the importance of these relatively new methods and that their many use cases belie their widespread research and use. However, they have not completely displaced the simpler machine learning architectures found in the yellow cluster, as they are an important pillar for research in industrial computing. It is particularly important to note that our analysis of the term and citation maps showed that two types of research have been prominent since the 2000s: Algorithms and Software and Industrial Hardware and Embedded Systems. The position of these clusters shifts significantly between the penultimate and most recent periods, suggesting that the state of the field is still changing rapidly and that the coming period may again show a shift in terms.

## 5. Conclusions

In this article, we have focused on examining and dissecting the state and development of the research field of industrial informatics. Industrial informatics is a rapidly growing field with many different players, resulting in a complex web of interconnected research areas. In such cases, bibliometric mapping can provide valuable information about the status and nature of fields or specialties. Its purpose is to map the relationships between different units of interest, such as journals, authors, research institutes, and phrases. To the best of our knowledge, this is the first attempt to examine the structure and evolution of research in industrial computing using a scientific mapping technique. After reviewing 1045 articles published between 1994 and 2023, we reached the following conclusions:

(i) Since 1995, the number of publications in the field of industrial informatics has increased rapidly.

(ii) *IEEE Transactions on Industrial Informatics* has been the most active journal for publishing research in industrial informatics.

(iii) China, Germany and Brazil dominate industrial informatics research.

(iv) Embedded systems, IoT, manufacturing, and automation are the research topics that characterize the current state of research in industrial computing.

This study is a first step in examining the structure of industrial informatics research. Future research could cover a variety of interrelated topics, such as analyzing collaboration among different actors (e.g., authors, organizations, and nations), characterizing differences in research among actors, assessing in more detail the research contributions of countries/universities to industrial informatics, and determining how subfields within industrial informatics have evolved over time.

**Supplementary Materials:** The following supporting information can be downloaded at: https://www.mdpi.com/article/10.3390/electronics12102238/s1. Table S1—VosViewer term citation map for the period 1994–2007, Table S2—VosViewer term citation map for the period 2008–2013, Table S3—VosViewer term citation map for the period 2014–2018, Table S4—VosViewer term citation map for the period 2019–2023.

**Author Contributions:** Conceptualization, A.I. and M.P.B.; formal analysis, D.H., A.I. and M.P.B.; methodology, A.I. and M.P.B.; validation, D.H. and M.P.B.; data curation, D.H. and A.I.; writing—original draft preparation, D.H., A.I. and M.P.B.; writing—review and editing, D.H. and A.I.; visualization, D.H. and A.I.; supervision, M.P.B.; funding acquisition, M.P.B. All authors have read and agreed to the published version of the manuscript.

**Funding:** This research received no external funding.

**Data Availability Statement:** The main analysis and figure data are available in the Supplementary Materials. Other data are available from the corresponding author upon reasonable request.

**Conflicts of Interest:** The authors declare no conflict of interest.

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
