# Peer review of "Industrial Informatics: Emerging Trends and Applications in the Era of Big Data and AI"

_electronics, doi:10.3390/electronics12102238_

Round 1

Reviewer 1 Report

The author has conducted a comprehensive review on industrial informatization, which is of great significance for the research of industrial informatization and also provides guidance for its development direction. Therefore,I recommend for publication.

Author Response

Thank you for your positive feedback. We appreciate your approval of the paper for publication. 

Reviewer 2 Report

Overall comments

The paper's strength is in its comprehensive review of Industrial Informatics research. Bibliometric analysis and text mining methods were used to investigate journals, conferences, authors, and nations involved in related scientific publications. The paper identified the topical groups of research in the field, providing readers with a clear understanding of current research and emerging trends.

Figures and Tables

Remove this note from all tables and figures: Source: Authors’ work (Of course it is your work).

Remove gray outline box from all figures.

Figure 4: Remove patterned fill.

Title

I would like to suggest that you pick any of these better titles:

1. "Exploring the Evolution of Industrial Informatics: A Bibliometric Analysis"

2. "Industrial Informatics: Emerging Trends and Applications in the Era of Big Data and AI"

3. A Comprehensive Review of Industrial Informatics using Bibliometric Analysis"

Abstract

These sentences are not needed in the abstract section. Send it to the introduction section:

"Industrial Informatics is a rapidly developing field that has emerged due to the progress of science, engineering, and information technology. Industrial Informatics primarily concerns knowledge-based automation of industrial engineering and manufacturing processes. Industrial Informatics is a set of methods and techniques that use information analysis, processing, and dissemination to improve industrial environment efficiency, effectiveness, stability, and safety. Industrial Informatics has evolved into one of the most important fields for intelligent manufacturing and management approaches. Industrial Informatics is not limited to the industrial sector; it is also used in knowledge-based businesses such as computer-based control systems, robots, vision systems, signal processing, and other data collection forms." 

Your first sentence should give general background of the topic.

Your second sentence should narrow direct to the main problem.

Add 2-3 sentences to report most important results.

Add 1 last sentence to suggest the implication of your study.

1. Introduction

Add one last paragraph to provide paper organizations/structure.

This sentence can be improved by rephrasing it to be more concise and precise:

"Industrial Informatics builds on a discourse on ICT design, innovation, and usage formulated in the informatics discipline."

This sentence is too long and contains too many ideas. It can be improved by breaking it up into smaller, more focused sentences:

"Due to its multidisciplinary nature, research in Industrial Informatics has attracted a great deal of attention from wildly different actors, and the breadth of Industrial Informatics has widened significantly from a supporting role for the design of manufacturing processes to usage throughout the entire product life cycles, including reusing, remanufacturing, maintenance, recycling and outsourcing."

This sentence can be improved by adding more focused context to make it clearer:

"The approach for creating and implementing the framework for such interconnected systems differs depending on each particular monitored physical system." 

2. Methodology

Remove period symbol:

1.077 documents (Should be 1077 or 1,077 documents)

1.045 documents (Should be 1077 or 1,077 documents)

"To construct the complex view of Industrial Informatics, we utilized bibliometric maps [22,23] created via VOSviewer, software specialized for quantifying and visualizing scientific bibliometric maps."

Suggested revision: To construct a comprehensive view of Industrial Informatics, we employed bibliometric maps [22, 23] generated using VOSviewer software, which is specialized in quantifying and visualizing scientific bibliometric maps.

"In term citation maps, the color assigned to a particular term shows its average citation impact."

Suggested revision: The color assigned to a term in term citation maps reflects its average citation impact.

3. Results

We retrieved 1.045 publications of various types from the Scopus database during a 29-year period.

Suggested revision: We retrieved 1,045 publications of various types from the Scopus database during a 29- year period.

In the abstract section you mentioned 30 years and not 29.

4. Discussion

Cut that long single paragraph into 2-3 shorter ones.

Add 3 limitations of your study.

Add 2 implications (theoretical and practical) of your study.

"This research interest is global, as 68 countries contributed to our dataset with at least one publication."

Suggested revision: "This research has a global interest, with at least one publication contributed by 68 countries in our dataset."

"During the time periods 1994–2007 and 2008–2013, the lack of terms on the map as well as their ambiguity prevented grouping into specific research areas."

The sentence can be made clearer. Suggested revision: "The lack of terms on the map and their ambiguity during the time periods of 1994–2007 and 2008–2013 hindered the grouping of specific research areas."

"Comparing the maps across the study periods allows for the observation of term emergence and disappearance, as well as term re-positioning within individual clusters, providing vital information on the dynamic of research subjects handled in a certain area map."

This sentence is too long and unclear. Suggested revision: "Comparing the maps across study periods allows us to observe term emergence, disappearance, and re-positioning within individual clusters, providing valuable insights into the dynamic nature of research subjects in a particular area map."

5. Conclusions

Add 2 more future work suggestions.

Author Response

Dear reviewer, thank you for your positive feedback and useful comments for the paper improvement. 

Comment

Response

Overall comments

The paper's strength is in its comprehensive review of Industrial Informatics research. Bibliometric analysis and text mining methods were used to investigate journals, conferences, authors, and nations involved in related scientific publications. The paper identified the topical groups of research in the field, providing readers with a clear understanding of current research and emerging trends.

Thank you for your positive feedback. We appreciate your approval of the paper for publication

Figures and Tables

Remove this note from all tables and figures: Source: Authors’ work (Of course it is your work).

Remove gray outline box from all figures.

Figure 4: Remove patterned fill.

We tweaked all figures and figure captions as suggested.  

Title

I would like to suggest that you pick any of these better titles:

1. "Exploring the Evolution of Industrial Informatics: A Bibliometric Analysis"

2. "Industrial Informatics: Emerging Trends and Applications in the Era of Big Data and AI"

3. A Comprehensive Review of Industrial Informatics using Bibliometric Analysis"

Thank you for the suggestions, we have changed our title accordingly.

Abstract

These sentences are not needed in the abstract section. Send it to the introduction section:

"Industrial Informatics is a rapidly developing field that has emerged due to the progress of science, engineering, and information technology. Industrial Informatics primarily concerns knowledge-based automation of industrial engineering and manufacturing processes. Industrial Informatics is a set of methods and techniques that use information analysis, processing, and dissemination to improve industrial environment efficiency, effectiveness, stability, and safety. Industrial Informatics has evolved into one of the most important fields for intelligent manufacturing and management approaches. Industrial Informatics is not limited to the industrial sector; it is also used in knowledge-based businesses such as computer-based control systems, robots, vision systems, signal processing, and other data collection forms." 

Your first sentence should give general background of the topic.

Your second sentence should narrow direct to the main problem.

Add 2-3 sentences to report most important results.

Add 1 last sentence to suggest the implication of your study.

Thank you for your detailed suggestions, they are excellent. We have rewritten the abstract and introduction to incorporate them.

1. Introduction

Add one last paragraph to provide paper organizations/structure.

This sentence can be improved by rephrasing it to be more concise and precise:

"Industrial Informatics builds on a discourse on ICT design, innovation, and usage formulated in the informatics discipline."

This sentence is too long and contains too many ideas. It can be improved by breaking it up into smaller, more focused sentences:

"Due to its multidisciplinary nature, research in Industrial Informatics has attracted a great deal of attention from wildly different actors, and the breadth of Industrial Informatics has widened significantly from a supporting role for the design of manufacturing processes to usage throughout the entire product life cycles, including reusing, remanufacturing, maintenance, recycling and outsourcing."

This sentence can be improved by adding more focused context to make it clearer:

"The approach for creating and implementing the framework for such interconnected systems differs depending on each particular monitored physical system." 

Thank you for your suggestions. We have rephrased the offending sentences and added an overview of the paper structure to the introduction.

2. Methodology

Remove period symbol:

1.077 documents (Should be 1077 or 1,077 documents)

1.045 documents (Should be 1077 or 1,077 documents)

"To construct the complex view of Industrial Informatics, we utilized bibliometric maps [22,23] created via VOSviewer, software specialized for quantifying and visualizing scientific bibliometric maps."

Suggested revision: To construct a comprehensive view of Industrial Informatics, we employed bibliometric maps [22, 23] generated using VOSviewer software, which is specialized in quantifying and visualizing scientific bibliometric maps.

"In term citation maps, the color assigned to a particular term shows its average citation impact."

Suggested revision: The color assigned to a term in term citation maps reflects its average citation impact.

We thank you for your proposed imporvements, we have incorporated them in the revised manuscript.

3. Results

We retrieved 1.045 publications of various types from the Scopus database during a 29-year period.

Suggested revision: We retrieved 1,045 publications of various types from the Scopus database during a 29- year period.

In the abstract section you mentioned 30 years and not 29.

Thank for pointing out the year error and the incorrectly formatted number, both have been corrected.

4. Discussion

Cut that long single paragraph into 2-3 shorter ones.

Add 3 limitations of your study.

Add 2 implications (theoretical and practical) of your study.

"This research interest is global, as 68 countries contributed to our dataset with at least one publication."

Suggested revision: "This research has a global interest, with at least one publication contributed by 68 countries in our dataset."

"During the time periods 1994–2007 and 2008–2013, the lack of terms on the map as well as their ambiguity prevented grouping into specific research areas."

The sentence can be made clearer. Suggested revision: "The lack of terms on the map and their ambiguity during the time periods of 1994–2007 and 2008–2013 hindered the grouping of specific research areas."

"Comparing the maps across the study periods allows for the observation of term emergence and disappearance, as well as term re-positioning within individual clusters, providing vital information on the dynamic of research subjects handled in a certain area map."

This sentence is too long and unclear. Suggested revision: "Comparing the maps across study periods allows us to observe term emergence, disappearance, and re-positioning within individual clusters, providing valuable insights into the dynamic nature of research subjects in a particular area map."

We thank you for you kind suggestions. We have rewritten the discussion to include limitations and implications, and cleared up the the longer sentences along the lines you suggested.

5. Conclusions

Add 2 more future work suggestions.

We have expanded the last paragraph of Conclusions to include more future reasedarch opportunities.

Reviewer 3 Report

The article analyzes publications on an important field - industrial informatics in a very long period of time (from 1994 to 2023). The article could have many readers.

Author Response

Comment

Response

The article analyzes publications on an important field - industrial informatics in a very long period of time (from 1994 to 2023). The article could have many readers.

Thank you for your positive feedback. We appreciate your approval of the paper for publication.

Reviewer 4 Report

I have no substantive or methodological comments. The structure of study is correct. The presentation of the research is clear.

In references, only four items from 2022,2023. Recomended updating of the lierature used.

Minor editorial remarks are marked in yellow in the text.

Author Response

I have no substantive or methodological comments. The structure of study is correct. The presentation of the research is clear.

In references, only four items from 2022,2023. Recomended updating of the lierature used.

Minor editorial remarks are marked in yellow in the text.

We thank you for the suggestions regarding citations and the editorial remarks.

We have added newer citations where appropriate and corrected the editorial errors.

Reviewer 5 Report

The paper shows good scientific merit but several discussions require further justification, as commented below.

1.   The figure captions of Figure 10 and Figure 11 are the same.

2.     Although the authors have built the term citation maps over the decades, the citation maps should be analyzed in detail. The representative studies in the application of Industrial Informatics should be elaborated. E.g. in figure 12, the related works about energy efficiency (DOI:10.1109/ISTM54910.2022.00014; DOI:10.1038/s41467-022-29320-6), fault detection (DOI:10.1016/j.compind.2021.103551; DOI: 10.3390/electronics9091547), digital storage(Doi :10.1109/MSEC.2018.2875877) should be cited.

3.     The author should improve the quality of all the figures and increase the text size appropriately.

4.     The paper needs careful editing by professional services or individuals. The authors should pay particular attention to English sentence structure to make the contents clear to understand.

Author Response

Comment

Response

The paper shows good scientific merit but several discussions require further justification, as commented below.

We thank you for your comments. We have incorporated your suggestions into the revised manuscript.

1.   The figure captions of Figure 10 and Figure 11 are the same.

Thank you for your comment, we have corrected the erroneous figure caption.

2.     Although the authors have built the term citation maps over the decades, the citation maps should be analyzed in detail. The representative studies in the application of Industrial Informatics should be elaborated. E.g. in figure 12, the related works about energy efficiency (DOI:10.1109/ISTM54910.2022.00014; DOI:10.1038/s41467-022-29320-6), fault detection (DOI:10.1016/j.compind.2021.103551; DOI: 10.3390/electronics9091547), digital storage(Doi :10.1109/MSEC.2018.2875877) should be cited.

Thank you for your suggestions. We have expanded the discussion sections of each time period and referenced papers as appropriate.  

3.     The author should improve the quality of all the figures and increase the text size appropriately.

We have increased the resolution of all figures and changed the font sizes where needed.

4.     The paper needs careful editing by professional services or individuals. The authors should pay particular attention to English sentence structure to make the contents clear to understand.

Thank you for your comment – we have professionally checked the paper and it should now be up to the highest standards.

Round 2

Reviewer 2 Report

Now I am satisfied with the corrections made.